# An Overview of the Enhanced Effects of Curcumin and Chemotherapeutic Agents in Combined Cancer Treatments

**DOI:** 10.3390/ijms241612587

**Published:** 2023-08-09

**Authors:** Nunzio Antonio Cacciola, Rossana Cuciniello, Gianluigi Daniele Petillo, Miriam Piccioni, Stefania Filosa, Stefania Crispi

**Affiliations:** 1Department of Veterinary Medicine and Animal Production, University of Naples Federico II, Via F. Delpino 1, 80137 Naples, Italy; nunzioantonio.cacciola@unina.it; 2Research Institute on Terrestrial Ecosystems (IRET), UOS Naples-Consiglio Nazionale delle Ricerche (CNR), Via Pietro Castellino 111, 80131 Naples, Italy; 3Institute of Biosciences and BioResources-UOS Naples CNR, Via P. Castellino, 111, 80131 Naples, Italy; rossana.cuciniello@ibbr.cnr.it (R.C.); miriampiccioni@gmail.com (M.P.); 4IRCCS Neuromed, 86077 Isernia, Italy; 5School of Medicine, Catholic University of the Sacred Heart, L.go A. Gemelli 8, 00168 Rome, Italy; gianluigi.petillo01@icatt.it

**Keywords:** curcumin, cancer, combined treatment, phytotherapy, cancer resistance, signalling pathway

## Abstract

Due to the progressive ageing of the human population, the number of cancer cases is increasing. For this reason, there is an urgent need for new treatments that can prolong the lives of cancer patients or ensure them a good quality of life. Although significant progress has been made in the treatment of cancer in recent years and the survival rate of patients is increasing, limitations in the use of conventional therapies include the frequent occurrence of side effects and the development of resistance to chemotherapeutic agents. These limitations are prompting researchers to investigate whether combining natural agents with conventional drugs could have a positive therapeutic effect in cancer treatment. Several natural bioactive compounds, especially polyphenols, have been shown to be effective against cancer progression and do not exert toxic effects on healthy tissues. Many studies have investigated the possibility of combining polyphenols with conventional drugs as a novel anticancer strategy. Indeed, this combination often has synergistic benefits that increase drug efficacy and reduce adverse side effects. In this review, we provide an overview of the studies describing the synergistic effects of curcumin, a polyphenol that has been shown to have extensive cytotoxic functions against cancer cells, including combined treatment. In particular, we have described the results of recent preclinical and clinical studies exploring the pleiotropic effects of curcumin in combination with standard drugs and the potential to consider it as a promising new tool for cancer therapy.

## 1. Introduction

Despite the discoveries made in recent years regarding the treatment of cancer, this disease still affects a large proportion of the world’s population. The American Cancer Society estimates that in 2022, 1.9 million new cases of cancer will have been diagnosed in the United States and 609,360 people will have died from cancer [1]. The main difficulty in treating cancer is related to one of its main characteristics, namely its complex genetic background and its ability to modify it, which leads to the circumvention of traditional pharmacological treatments. This phenomenon often arises as a consequence of monotherapy, which causes the selection of random DNA mutations responsible for drug resistance [2]. Another limitation of standard chemotherapy for cancer is the occurrence of serious side effects that often limit the ability to treat this disease. These include cardiotoxicity [3], hepatotoxicity [4], nephrotoxicity [5] and ototoxicity [6].

In recent decades, research interest has increasingly focused on the development of novel molecules to cure cancer. Most alternative cancer therapies are based on the use of bioactive molecules derived from natural sources that are able to significantly reduce tumour progression and improve cure and survival [7]. In this context, several natural molecules have been reported to be non-toxic and effective against different types of cancer [8].

An increasing number of studies on the efficacy of natural phytochemicals such as polyphenols have demonstrated that these molecules are able to significantly block or attenuate cancer growth. These studies have also shown the possibility of using these molecules in combination with conventional anticancer drugs, thus reducing the limitations associated with side effects or multidrug resistance (MDR) [9]. Interestingly, bioactive plant compounds often show a synergistic effect when used in combination with standard drugs. This leads to an increased inhibition of tumour growth and a chemoprotective effect on the healthy cells without obvious side effects [10]. The advantage of combination treatments is the possibility of increasing the efficacy of drugs and reducing their dose, which leads to a reduction in adverse effects. When designing combinations of molecules to be used as multitargeted anticancer drugs, it is important to take into account the numerous mutations specific to each cancer type, as these mutations would become the targets of treatment and prevent the development of drug resistance.

The National Cancer Institute (NCI) has recently launched a new program for the discovery of bioactive natural molecules [11] and it has already identified about 35 plant compounds with cancer-preventive properties, mainly due to their antioxidant activity [12]. Among them, epigallocatechin-3-gallate (EGCG) [13], cannabinoids [14], γ-tocotrienol (γ-T3) [15], quercetin [16] and curcumin [17] are the best-known natural molecules with stronger efficacy. The anticancer effects of phytochemicals have been demonstrated by a large number of studies that have shown their ability to coordinate the expression of genes and proteins involved in carcinogenesis [18,19]. It has also been shown that synergism can occur between phytochemicals and traditional medicines, enhancing the anticancer effect [20].

Curcumin, a yellow pigment of the spice turmeric *Curcuma longa* Linn [21], is one of the most studied phytochemicals used as anticancer molecules. This polyphenol has been used in traditional medicine in Asia since ancient times and it has been shown to have extensive cytotoxic effects against cancer cells [22]. In addition, curcumin has also shown beneficial and protective functions in acute myocardial infarction and cardiovascular diseases [23], chronic atherosclerosis [24,25], neurodegenerative diseases [26] and autoimmune diseases [27]. These activities are not covered in this review.

This review gives a general overview of the main anticancer properties of curcumin and, in particular, the synergism seen when it is used in combination with standard chemotherapeutic agents. It also presents the results of preclinical studies and recent clinical trials that demonstrate the pleiotropic action of curcumin, through which it exerts the main anticancer effects.

## 2. Curcumin

### 2.1. Chemical Structure and Anticancer Effects

Curcumin belongs to the chemical class of polyphenols, molecules characterised by a polyphenolic structure containing one or more hydroxyl groups and at least one aromatic ring. Curcumin is the major constituent of the rhizomes of *Curcuma longa* L., commonly known as turmeric, and it has been one of the most studied herbal remedies, attracting strong pharmaceutical attention due to its diverse biological activities [21].

The biological properties of curcumin are related to its chemical structure (diferuloylmethane). Turmeric contains about 2–9% curcuminoids. Commercial turmeric extracts contain about 70–75% curcumin, 20% demethoxycurcumin and 5% bisdemethoxycurcumin (Figure 1).

Chemically, curcumin is a β-diketone α-β-unsaturated ferulic acid. The bioavailability of native curcumin is generally poor, and it is rapidly metabolised after oral ingestion, with blood levels being extremely low. However, the active metabolites of curcumin can be formed by specific enzymes of the gut microbiota [26]. The safety of curcumin and turmeric products has been confirmed by the Food and Drug Administration (FDA), the Food and Agriculture Organisation (FAO) and the World Health Organisation (WHO) [28].

Interest in curcumin has gradually increased after several studies demonstrated its chemotherapeutic and chemopreventive effects. Several studies have reported that curcumin is able to affect gene expression and trigger various signalling mediators such as *NF-κB*, *COX-2*, *AP-1*, *EGFR*, *MMP9* and *PKC*, which are involved in several signalling pathways such as angiogenesis, survival, metastasis and proliferation [29,30]. Curcumin has been reported to exert significant immunomodulatory activity and protect the immune system from cancer-related suppression [31,32].

Since curcumin is able to modulate gene expression, in vitro studies have been conducted to investigate transcriptional modulation associated with the use of this phytochemical. Transcriptomic analyses performed after curcumin treatment in various cancer cells showed specific modulation of gene expression [33,34]. Further analysis of microRNA expression showed that curcumin can also modulate various non-coding genes in different cellular contexts, such as miR-122 and miRNA-199a* in human pancreatic cancer or miR-132-3p, miR-183-5p, miR-124-3p, miR-215, miR-192-5p and miR-194-5p in non-small cell lung cancer [35,36].

In addition, other studies have shown that curcumin can influence the epigenomic landscape, as it is able to induce specific methylation changes that are not determined by a direct effect on DNA methyltransferases but are a consequence of curcumin’s direct effect on modulating gene expression [37].

As for the cytotoxic effects of curcumin in cancer, this ability is closely linked to the modulation of signalling pathways related to cancer, such as cell cycle, apoptosis, and resistance to multiple drugs. For example, curcumin has been reported to prevent colon cancer proliferation by blocking the cell cycle and accelerating apoptosis. It exerts this effect on thymidylate synthase and its transcription factor E2F-1. This effect resulted in cell cycle inhibition via the downregulation of NFκ-B and other survival pathways [38]. In addition, curcumin downregulated the kinase CDK2, leading to the G1 cell cycle [39]. In human colon cancer cells, curcumin significantly inhibited cell growth. It also triggered apoptosis via a mitochondria-mediated pathway. Curcumin induced the release of cytochrome c, significantly increased BAX and p53, and showed a marked reduction in BCL-2 and survivin in colorectal carcinoma LoVo cells [40]. Moreover, curcumin was reported to significantly decrease the expression of cyclin D and inhibit the activity of p21-activated kinase1 (PAK1), leading to the suppression of gastric cancer cell proliferation and invasion [41]. In fact, it acted on cell cycle arrest in the G2/M phase in AGS cells via decreasing cyclin D1 and increasing cyclin B1 in a dose-dependent manner [42].

In addition, curcumin can induce apoptosis by triggering both the extrinsic and intrinsic pathways, acting on TNFR (Tumor Necrosis Factor Receptor) death signalling [43] and the p53 death signalling network [17,44], respectively. Of note, curcumin can also affect the expression of anti-apoptotic and pro-survival proteins (such as BCL-2, AKT and c-JUN) and induce the expression of pro-apoptotic genes (BAX, PPARγ and p21) [17,44,45].

Curcumin is able to prevent multidrug resistance by specifically reducing the expression of proteins involved in drug efflux [46]. In particular, curcumin and its derivatives can interfere with the activity of efflux drug transporters of the ATP-binding cassette family (ABC), including P-glycoprotein, multidrug resistance protein (MRP) and breast cancer resistant protein (BCRP), which act as ATP-dependent efflux pumps that actively regulate the excretion of a number of drugs and limit their systemic bioavailability [47,48]. Curcumin has also been found to affect the activity of phase I biotransformation enzymes such as cytochrome P450 (CYP) 3A4 (CYP3A4) [49], which catalyses the metabolism of about half of all drugs marketed in the USA [50]. In healthy Japanese volunteers, curcumin (2 g) was found to increase the plasma concentration of sulfasalazine after administration of a therapeutic dose (2 g) of the anti-inflammatory drug sulfasalazine (salazopyrine, azulfidine) [51].

Since curcumin is able to modulate genes involved in key signalling pathways that play a critical role in cancer and reverse drug resistance induced by conventional chemotherapeutic agents, it is a remarkable candidate to be considered in combination anticancer treatment regimens.

### 2.2. Curcumin Combined Treatments Affect Different Pathways

The major limitation in the use of conventional drugs in cancer treatment is the adverse effects that can significantly affect the outcome of treatment. Together with the development of drug resistance, these limitations are forcing researchers to look for possible solutions to realise the full potential of chemotherapeutic agents. A new strategy that has been considered recently is combined treatment with natural molecules such as polyphenols, including curcumin. These treatments often ameliorate the specific effects of the drug. In addition, the wide range of functions performed leads to an increase in the effectiveness of the treatment by attenuating the adverse effects and reducing drug resistance. Numerous preclinical studies have shown over time that curcumin alone, in different concentrations or formulations, is capable of producing significant anticancer effects in vitro as well as in vivo in various types of cancer (Figure 2). Depending on the combination of active ingredients, curcumin acts by modulating different molecular signalling pathways.

Here, we report some examples of the molecular pathways modulated by the combination of curcumin with the most studied chemotherapeutic agents: cisplatin, doxorubicin, 5-fluorouracil (5-FU) and gefitinib. Cisplatin is the main representative of platinum-alkylating compounds and is widely used to treat many solid cancers due to its efficacy and broad spectrum of activity [52]. Cisplatin binds genomic or mitochondrial DNA, determining DNA lesions that lead to disruption of DNA replication and protein production and induce apoptosis or necrosis. The potential of cisplatin is often limited by resistance caused by mechanisms such as inactivation of the drug, reduced accumulation, DNA repair and deregulation of apoptotic signals [53].

In a recent study, it has been shown that curcumin induces a strong ER (Endoplasmic Reticulum) stress in non-small cell lung cancer (NSCLC) cells, making them more sensitive to the effects of cisplatin. In particular, the results suggest that cisplatin resistance—which is related to ER stress—can be reversed by curcumin as it acts directly on key proteins of the ER stress pathway such as GRP78, XBP-1, ATF6 and DDIT3 [54].

Curcumin has been used in combination with cisplatin for hepatocellular carcinoma. In this study, both compounds were introduced into cells after being co-loaded into liposomes and were able to determine apoptosis in both mouse hepatoma cells (H22) and human hepatocarcinoma cells (HepG2) and their derived xenografts. The cytotoxic effect induced by the combined treatment with curcumin and cisplatin was associated with a higher production of intracellular ROS. The higher production of ROS was shown to determine the modulation of several apoptosis-related genes such as *Bcl-2*, *Sp1*, *P-Erk1/2*, *p53*, *caspase-3* and *Bax*. The results showed that the reduced presence of ROS in both cell lines affects the expression of these genes, confirming that the synergistic effect of curcumin and cisplatin determines the increase in ROS levels responsible for the anticancer activity [55].

Interestingly, in colorectal cancer, one study reported that curcumin reverses cisplatin resistance by acting on long non-coding RNAs. Specifically, the study demonstrated curcumin’s ability to decrease the expression of the LncRNA KCNQ1OT1, an oncogene upregulated in many cancers. KCNQ1OT1 impairs the expression of the miR-497/*Bcl-2* axis and makes colorectal cancer cells resistant to cisplatin by promoting survival, proliferation and invasion. Downregulation of KCNQ1OT1 by curcumin results both in vitro and in vivo in cell viability and tumour growth decrease [56].

Doxorubicin is another standard drug that is often used to treat cancer. It belongs to the class of anthracycline antibiotics and is used to treat solid and haematological cancers. Doxorubicin exerts its function by combining intercalation into DNA and inhibition of topoisomerase II, causing cell death or cell cycle block [57].

Adverse effects of doxorubicin may occur during treatment but also continue for many years after treatment has completed. Long-term treatments based on the use of this drug are often limited by the development of drug resistance.

The development of drug resistance may be due to the overexpression of ATP-binding cassette transporters (ABC). ABC transporters often consist of different subunits and one or two of them are transmembrane proteins. They also contain ATPase subunits that use the energy of adenosine triphosphate (ATP) for the uptake or export of substrates across membranes [58]. It has been observed that doxorubicin-resistant breast cancer cells are characterised by the overexpression of ABC subfamily B member 4 (ABCB4), which inhibits the accumulation of drugs by expelling them from the cell [59].

Constitutive activation of NF-κB is another mechanism related to the development and progression of drug resistance [60]. In this case, NF-κB is translocated to the nucleus as a result of increased degradation of IκB, a protein that masks NF-κB nuclear localisation signals, where it binds the transcriptional coactivator p300 histone acetyltransferase (HAT), whose function is critical for gene regulation. This in turn induces the transcription of *Bcl-2* [61].

The NF-κB signalling pathway is an important signalling pathway that is affected by curcumin [62]. This signalling pathway is often upregulated in breast cancer, resulting in the hyperactivation of anti-apoptotic proteins that cause greater resistance to doxorubicin treatments. Curcumin has been shown to be able to sensitise breast cancer epithelial cells to doxorubicin in vitro and in a breast cancer mouse model in vivo, both of which are resistant to treatment. Specifically, curcumin was shown to induce the cleavage of p300 from NF-κB, allowing it to bind p53. The p53-p300 and PML-SMAR1 cross-talk leads to the activation of the transcriptional functions of p53, resulting in the transactivation of apoptotic factors such as PUMA, NOXA and BAX and the chemosensitisation of cancer-resistant cells [63].

5-FU is one of the first chemotherapeutic agents discovered and used. It has been used for over 50 years to treat various types of cancer, including stomach, breast, liver and prostate cancer. It has been reported that the anticancer effect of 5-FU increases with increasing dosage. Unfortunately, the cytotoxicity of 5-FU also affects normal cells, resulting in excessive toxicity in cancer patients. 5-FU causes severe adverse effects such as cardiotoxicity, hepatotoxicity and cognitive problems. 5-FU belongs to the class of anti-metabolites that act by inhibiting essential biosynthetic processes or by incorporation into DNA and RNA, inhibiting their normal function by blocking repair mechanisms and inducing cell death [64].

In particular, 5-FU causes misincorporation of fluoronucleotides into RNA and DNA and inhibits the nucleotide synthesis enzyme thymidylate synthase (TS). 5-FU is normally used to treat a number of cancers, including stomach, oesophagus, breast and pancreatic cancer.

A very recent study described that a combined treatment based on curcumin and 5-FU can overcome the development of resistance. Cancer resistance to 5-FU is often characterised by the upregulation of nicotinamide N-methyltransferase (NNMT), which is associated with increased aggressiveness. The importance of NNMT in cancer progression makes it an interesting therapeutic target. Another study in colorectal cancer reported that curcumin acts synergistically with 5-FU both in vitro and in vivo. Again, the combined treatment was able to affect cancer resistance by reducing NNMT expression at both the mRNA and protein levels as a result of the downregulation of p-STAT3, a well-known upstream regulator of NNMT. In addition, a ROS-induced cell cycle arrest was observed [65].

## 3. The Combination of Curcumin and Chemotherapy Drugs in Cancer Therapy

The chemotherapeutic agents described in the previous section have been used in combination with curcumin for different types of cancer (Table 1). The following are the main results based on their use in different types of cancer.

### 3.1. Curcumin and Cisplatin Combined Treatment

The combined effect of curcumin and cisplatin in cancer has been extensively studied, and lung cancer is possibly the best studied cancer model for this drug combination. One of the first pioneering studies by Ichiki et al. [66] showed that this treatment strongly inhibited growth and prolonged survival in an orthotopic mouse model of lung cancer. In addition, curcumin was shown to affect the expression of activator protein-1 (AP-1), a protein associated with lymphatic metastasis in lung cancer patients. Recently, another in vivo study using both ectopic and orthotopic lung tumour mouse models confirmed the efficacy of this treatment in non-small cell lung cancer (NSCLC) cells and showed that curcumin potentiates the anticancer effect of cisplatin, which is associated with the down-regulation of the expression of COX-2, p-ERK1/2 and EGFR [67].

Curcumin in combination with cisplatin has been shown to reduce ovarian cancer cell cycle and increase apoptosis in a dose- and time-dependent manner [68]. In a later study, it was reported that synergistic effects were observed mainly when the two agents were administered separately and with a time interval of a few hours. It was postulated that curcumin improves the uptake of cisplatin and increases cisplatin DNA adducts and thus apoptosis [69].

The efficacy of this drug combination has been studied in breast cancer, one of the most complex diseases, which is considered a multifactorial disease that has multiple targets. A study by Zou et al. [70] analysed the role of curcumin in reversing cisplatin resistance in breast cancer by investigating the role of FEN1, an endonuclease that stimulates base excision repair, overexpression of which correlates with cancer development and promotes cisplatin resistance in breast cancer cells. The results showed that curcumin downregulated the expression of FEN1 in a dose-dependent manner both in vitro and in vivo, suggesting that FEN1 could be a potential therapeutic target to alleviate cisplatin resistance in breast cancer. Another study showed that curcumin pretreatment enhanced the anticancer effect of cisplatin while reducing its nephrotoxicity. This treatment effectively reduced tumour growth in vitro and in vivo without affecting kidney growth. The combined treatment resulted in enhanced expression of PPAR-γ and reduced expression of BDNF. Interestingly, activation of PPAR-γ leads to an anticancer effect in experimental models of breast cancer, and high BDNF levels have been associated with breast cancer development and resistance [71]. Furthermore, in a rat model of breast cancer, pretreatment with curcumin together with cisplatin was reported to synergistically enhance its anticancer effect while reducing nephrotoxicity [72]. Recently, an in vivo study showed that treatment with curcumin followed by cisplatin resulted in complete regression of tumour mass in induced breast cancer. In addition, curcumin had a positive effect on the expression of *Par4*, a tumour suppressor gene whose decreased expression is associated with poor prognosis [73].

Curcumin-induced sensitisation to cisplatin has also been observed in other cancer cell models such as bladder cancer, glioblastoma, colorectal cancer and laryngeal cancer. In bladder cancer, the development of resistance to cisplatin is a major obstacle to treatment efficacy. It is known that oxidative stress induced by cisplatin contributes to its cytotoxic effect and that increased antioxidant mechanisms of cancer cells attenuate cisplatin-induced apoptosis [74]. A study conducted both in vitro and in vivo with various bladder cancer cells investigated whether the pro-oxidant activities of curcumin could increase the efficacy of cisplatin. The results showed that curcumin induced apoptosis through the ROS-mediated activation of ERK1/2 [75]. ERK belongs to the MAPK superfamily and is known for its ability to modulate cell survival in response to external stimuli. Increasing its expression can promote apoptosis in certain environments. It has been reported that in bladder cancer cells, combined treatment with curcumin and low doses of standard clinical drugs, including cisplatin, reversed drug resistance, inhibited cell proliferation and induced apoptosis by suppressing the expression of HER2, an oncogenic protein involved in bladder cancer [76].

Glioblastoma is a primary brain tumour that is very aggressive and difficult to treat. Several studies have reported that curcumin can induce apoptosis in this tumour by activating TRAIL [77], a TNF superfamily cytokine that mainly activates the extrinsic pathway of apoptosis, or by inhibiting the expression of the metalloprotease MMP9 [78]. Recently, a study on human and rat glioblastoma cell lines showed that curcumin can overcome their radioresistance and chemoresistance. Also in this case, curcumin was able to sensitise glioma cells to cisplatin by decreasing the expression of BCL-2 and members of the IAP family, as well as DNA repair enzymes [79]. Another preclinical study reported that curcumin was able to inhibit the expression of NF-κB, which is a target for the induction of cell death in this cancer, in synergy with cisplatin. In addition, cisplatin-resistant cells were sensitive to NF-κB inhibitors, so that the combination of cisplatin and NF-κB inhibitors such as curcumin overcame cisplatin resistance [80].

Several recent studies suggest that the combined use of curcumin with standard chemotherapeutic agents may also be considered an effective approach for the treatment of laryngeal carcinoma. In an in vitro study, curcumin was shown to cause cell death in human laryngeal cancer cells by activating the TRPM2 (Transient receptor potential melastatin 2) channel. Furthermore, curcumin increased intracellular and mitochondrial oxidative effects and decreased cisplatin resistance [81]. Another study investigated the anticancer effect of treatment with a combination of curcumin and cisplatin on CD133+ cancer stem cells responsible for drug resistance in laryngeal cancer. The results showed that the combined treatment increased the sensitivity of cells to cisplatin by suppressing ATP-binding cassette sub family G member 2 (ABCG2), which is involved in chemoresistance [82]. Curcumin has been reported to alleviate cisplatin-induced bladder dysfunction, improving the quality of life of cisplatin-treated patients. This result may be attributed to the antioxidant effect of curcumin and its ability to activate the NRF2 protein [83].

### 3.2. Curcumin and Doxorubicin Combined Treatment

Doxorubicin is the most commonly used chemotherapeutic agent for breast cancer. As with other drugs, the frequent development of doxorubicin resistance limits the long-term treatment of patients. Combined treatment with curcumin and doxorubicin was reported to reverse resistance in several human breast cancer cells. Specifically, curcumin inhibited the transport function of ABCB4, thereby reducing doxorubicin resistance. ABCB4 is a member of the ABC transporter family that is overexpressed in cancer cells and plays a key role in drug resistance [104]. Recently, in vitro and in vivo studies have shown that combined treatments with curcumin and doxorubicin showed better anticancer activity against breast cancer when the two compounds were co-administered with folic acid-modified nanoparticles [105] or a complex polymer-based nanomicellar system [106].

The development of multidrug resistance (MDR) is the main obstacle to treatment efficacy in leukaemia cancer as well. Several studies have shown that curcumin potentiates the effect of chemotherapeutic agents such as doxorubicin against leukaemia cells. In acute lymphoblastic leukaemia (ALL), this effect is achieved through the activation of caspase 3 and the downregulation of oxidative stress [84], while in acute myeloid leukaemia (AML), the efficacy of this combined treatment is related to the inhibition of the FLT3 protein, a leukaemia-related protein marker overexpressed on the cell surface of leukaemic cells [85]. In contrast, in chronic myeloid leukaemia (CML), this treatment was able to reverse the effects of drug resistance by downregulating the expression of P-glycoprotein (P-gp) and S100A8 and increasing doxorubicin-induced apoptosis [86]. Combined treatments for hepatocarcinoma are very limited. Two recent studies developed a co-delivery system based on the use of lipid nanoparticles or pH-sensitive polymeric nanoparticles to analyse the anticancer efficacy of doxorubicin and curcumin in vitro and in vivo. These studies showed that this system was able to increase anticancer efficacy and reverse MDR both in vitro and in vivo [87,88].

Curcumin has been shown as a potential anticancer agent against glioblastoma and its efficacy in potentiating the anticancer effect of doxorubicin was studied in different human glioma cell lines. The results showed that a 12 h pretreatment with curcumin followed by a 48 h co-incubation with doxorubicin potentiated the effect of doxorubicin and increased apoptosis. Indeed, curcumin was reported to sensitise glioma cells to doxorubicin by inhibiting the expression of BCL-2 and IAP family members and DNA repair enzymes [80].

Doxorubicin has limited therapeutic efficacy in gastric cancer [90]. However, the combined use of curcumin and doxorubicin is able to reverse chemoresistance by downregulating *NF-κB* and the anti-apoptotic genes *Bcl-2* and *Bcl*-*xL* [89]. Recently, it has been reported that treatment with curcumin and doxorubicin significantly reduces the cancer properties of gastric cancer cells, including viability, tumour spheroid formation, migration and invasion, and increases apoptosis [90].

The efficacy of curcumin and doxorubicin treatment in neuroblastoma was analysed in vitro using a 3D tumour spheroid culture as a cellular model of human neuroblastoma [91]. The results showed that the combined treatment inhibited cell migration and that pretreatment with curcumin enhanced the anticancer effect of doxorubicin by enhancing apoptosis [91].

### 3.3. Curcumin and 5-Fluorouracil (5-FU) Combined Treatment

Resistance to chemotherapy is a major cause of mortality in colorectal cancer. In this tumour, chemoresistance is primarily associated with a subset of cancer cells that undergo epithelial–mesenchymal transition. The efficacy of curcumin and 5-FU in combination was studied in colorectal cancer resistant to 5-FU. The results showed that curcumin enhanced the effect of 5-FU both in vitro and in vivo. In addition, curcumin was able to block the epithelial–mesenchymal transition by acting on the expression of specific genes that are key mediators of cancer stem cells [96]. The chemosensitising effect of curcumin on 5-FU in this cancer was also reported in another study based on the use of colon cancer cells encapsulated in alginate, which can proliferate in 3D colonospheres in an in vivo-like phenotype. Also in this study, curcumin was reported to enhance the efficacy of 5-FU, resulting in a reduction in cancer properties, along with the downregulation of NF-κB activation and NF-κB-regulated genes [93], resulting in a synergistic effect of the two compounds on colon cancer cells. More recently, these results were confirmed in vivo using a mouse model of colorectal cancer [94]. Although 5-FU is commonly used in the treatment of breast cancer, few studies have examined the effectiveness of 5-FU and curcumin in this cancer. Again, these studies suggest that combined treatment leads to synergistic improvements in preventing cancer cell proliferation and protects against 5-FU-induced cytotoxicity. In addition, pretreatment with curcumin causes the downregulation of thymidylate synthase (TS), an enzyme critical for de novo synthesis of DNA and the main target of 5-FU [95].

The efficacy of curcumin in enhancing the anticancer effect of 5-FU was investigated also in liver cancer. Using an ex vivo model derived from human colorectal liver metastases, it was shown that the addition of curcumin to 5-FU enhanced the overall anticancer effect and demonstrated the ability to target cancer stem-like cells [96]. Recently, these results were confirmed in vivo in a mouse model indicating that this treatment effectively inhibited liver cancer progression. The combined treatment also exerted a synergistic effect in suppressing tumour growth by decreasing AKT activity, thereby affecting the PTEN/PI3K/AKT pathway [97].

Squamous cell carcinoma of the oesophagus is a very aggressive cancer with poor prognosis and rapid progression. It is known that NF-κB activation plays an important role in the survival of these cancer cells and that curcumin downregulates this signalling pathway. Although further studies are needed, it has been reported that combined treatment with curcumin and 5-FU affects cell viability and cancer stem cell characteristics in this type of cancer [98,99].

5-FU is a drug commonly used to treat stomach cancer, but many patients develop resistance to it. Combining curcumin with 5-FU could increase chemotherapeutic efficacy in this cancer while reducing its toxicity both in vitro and in vivo. The effects might be related to the inhibition of the expression of the proteins COX-2 and NF-κB, which play a key role in the development and progression of gastric cancer [100]. In addition, curcumin prevents the formation of cancer-associated fibroblasts, which are responsible for the induction of chemoresistance in gastric cancer [101].

### 3.4. Curcumin and Gefitinib Combined Treatment

Gefitinib, the first selective inhibitor of the EGFR tyrosine kinase domain, is a widely used chemotherapeutic agent in various cancers. In lung cancer, its efficacy has been reported to be enhanced when combined with curcumin. In non-small cell lung cancer, this combined treatment enhanced the inhibitory effect of gefitinib on several primary cell lines that were resistant to the drug. At the molecular level, the effect was achieved by downregulating EGFR and blocking the interaction of Sp1 and HADC1 [102]. More recently, the increased efficacy of this treatment in this cancer has been reported through the use of a liposome-based co-delivery system [107].

All of these studies have shown that combined cancer treatments increase overall therapeutic efficacy compared to monotherapy. The combined use of standard drugs with specific natural molecules such as curcumin leads to increased sensitisation of cancer cells by inhibiting resistance mechanisms and activating cell death pathways.

## 4. Combined Treatments in Clinical Trials

As described earlier, numerous preclinical studies have shown that the combined treatment with curcumin and conventional drugs is of great benefit in the treatment of cancer. Therefore, clinical trials were subsequently conducted in different types of cancer, mainly to investigate the pharmacokinetics of the simultaneous administration of curcumin and standard drugs. In addition, it is important in these studies to investigate the possible interactions between the molecules selected for combined treatment, as natural molecules are usually taken as dietary supplements.

A search of the “clinicaltrials.gov” database using the terms “cancer” (condition) and “curcumin” (other terms) lists a total of 79 clinical trials, 13 of which included curcumin and various traditional medicines (Table 2). The completed trials are listed below.

One clinical trial was designed to evaluate the safety and tolerability of immunotherapy with bevacizumab, the active ingredient in Avastin, and chemotherapy with FOLFIRI (folinic acid, 5-FU and irinotecan) in combination with curcumin. In this trial, patients received a daily oral curcumin-containing supplement in the form of a ginsenoside-modified nanostructured lipid carrier with curcumin (G-NLC), a formulation with improved solubility and oral bioavailability [114,115]. Long-term survival and adverse events were evaluated over time. This study showed that the combination of curcumin and FOLFIRI had acceptable tolerability and safety. Although the patients who participated in the study had long-term survival comparable to the control group, compliance with chemotherapy was improved by curcumin. Despite several limitations due to the small population included, this study is noteworthy because it showed that curcumin may be associated with improved quality of life for patients with metastatic colorectal cancer when combined with current chemotherapeutic agents (NCT02439385) [108].

Another recently completed clinical trial used curcumin as part of an immunomodulatory cocktail containing vitamin D, aspirin, cyclophosphamide and lansoprazole. This cocktail was administered in combination with pembrolizumab and radiotherapy in patients with advanced and/or refractory cervical or endometrial cancer selected from “pre-treated persistent, recurrent, metastatic”. The study hypothesised that the combination of PD-1 (Programmed Death 1) with radiotherapy, immunomodulators and curcumin, which has radiosensitising and anti-inflammatory properties, could lead to clinical responses in patients. Unfortunately, despite promising preclinical data, this trial did not achieve significant results of clinical activity (NCT03192059) [109,110].

In patients with advanced pancreatic cancer, a phase 2 trial of gemcitabine and oral curcumin at a fixed dose was completed. The hypothesis of the study was that curcumin, as a natural agent with a potent antiproliferative effect, could improve the efficacy of the standard chemotherapy gemcitabine. The results showed that the combination treatment was not feasible, with about 30% of patients discontinuing curcumin due to gastrointestinal toxicities. Further studies are needed to investigate whether a different formulation of curcumin could improve the effect of chemotherapy in patients with pancreatic cancer (NCT00192842) [111]. A commercial standardised curcumin extract (C3 complex, Sabinsa) was combined with FOLFOX (folinic acid, 5-FU and oxaliplatin), a standard oxaliplatin-based chemotherapy, and administered in a multiphase study to patients with colorectal metastases. Preliminary experiments were performed on spheroids obtained from xenografts or explant cultures from five different patients with colorectal liver metastases. The results showed that the treatment was able to reduce the viability of the spheroids and downregulate the expression of pluripotent stem cell markers such as Oct3-4, AFP and HNF/FoxA2, and subsequently also Nanog, Otx2 and VEGFR2. Indeed, the combination of oxaliplatin and curcumin was able to significantly reduce proliferation and induce apoptosis. Based on these data, a phase I clinical trial was designed to evaluate the safety and tolerability of the combined treatment. A dose-escalation study was conducted, starting with a concentration of 500 mg, which was increased to 1 g and finally to 2 g of curcumin in the absence of adverse effects in combination with chemotherapy. No adverse effects occurred in about 80% of the patients [96]. Subsequently, a phase II trial assessed the beneficial effects on side effects caused by FOLFOX, as well as the overall improvements in patients’ disease response and survival. This clinical trial assessed that the combination of curcumin with FOLFOX chemotherapy is safe and tolerable and has the potential to benefit cancer patients (NCT01490996) [96,112,113].

A recently completed phase I study in patients with advanced colorectal cancer was designed to investigate the effect of curcumin on dose-limiting toxicity and the pharmacokinetics of irinotecan. Irinotecan is a camptothecin-derived drug used to treat many solid tumours such as lung, pancreatic and colorectal cancer. This drug has a complex metabolism. It is converted into an active metabolite called SN-38, which is responsible for inhibiting topoisomerase II and subsequently blocking DNA replication. SN-38 in its inactive form is converted to the inactive SN-38G via glucuronidation by a UDP-glucuronosyltransferase [116]. Since curcumin is able to inhibit the enzyme UDP-glucuronyltransferases, it could stabilise the active metabolite SN-38 and modulate its pharmacokinetics. The formulation used was a curcumin–phosphatidylcholine complex (PC, Meriva), which was primarily tested in a dose-escalation study with four different concentrations simultaneously with the administration of a fixed dose of irinotecan due to its better absorption compared to other formulations. Most participants experienced at least one adverse effect, mainly nausea. The pharmacokinetics of the combined administration of irinotecan and the higher dose of curcumin were evaluated using pharmacokinetic parameters such as AUC (area under the curve), Cmax (maximum concentration) and Tmax (time to maximum concentration). The results showed that curcumin did not alter the pharmacokinetics of irinotecan (NCT01859858) [117].

We do not discuss the other clinical trials reported in the database because they have unknown statuses, have been withdrawn or have been discontinued. The different results obtained in clinical trials often reflect the use of different curcumin preparations and formulations, which have different bioactivities.

## 5. Curcumin: Benefits and Limitations

The numerous preclinical studies reporting combined treatments with curcumin have shown different benefits since it is able to affect cancer pathways in different ways [118,119,120]. These results have increased interest in the use of curcumin in phytotherapy, and so it has been considered in the abovementioned clinical trials. Curcumin has shown a very promising safety profile [121]. According to reports by JECFA (Joint Expert Committee on Food Additives of the United Nations and the World Health Organisation) and EFSA (European Food Safety Authority), the acceptable daily intake (ADI) of curcumin is 0–3 mg/kg body weight [122]. Several studies in healthy volunteers have confirmed the safety and efficacy of curcumin. Despite its safety profile, limited side effects have been reported. For example, seven volunteers who received 500–12,000 mg in a dose-response study and were observed for 72 h experienced diarrhoea, headaches, skin rashes and yellow stools [123]. In another study, some subjects receiving 0.45–3.6 g curcumin per day for a period of one to four months reported nausea and diarrhoea, as well as an increase in serum levels of alkaline phosphatase and lactate dehydrogenase [124]. Despite the numerous applications of curcumin in in vitro or in vivo studies and clinical trials, there are some important limitations that need to be considered. First, curcumin is characterised by physicochemical properties that limit its use, such as low bioavailability and solubility or possible interaction with other compounds such as oxygen and metal ions. [125]. To overcome these issues, various nano-formulations or the use of nanoparticle-loaded drug delivery systems that contain curcumin have recently been developed to ensure complete absorption and increased efficacy [126]. However, these solutions are not enough to declare the use of curcumin as free from limitations and adverse effects, as there are still limitations that need resolution, such as potential adverse effects of curcumin derivatives that are highly absorbed [127]. Another promising approach is the simultaneous administration of curcumin with piperine, an alkaloid from black pepper and long pepper. Piperine significantly increases the bioavailability of curcumin—up to 2000%—by preventing its metabolism [128]. What complicates the resolution of these issues is primarily the lack of uniformity across all studies and trials dealing with curcumin, as the formulations used often differ greatly. Thus, data from different studies report beneficial effects associated with a particular formulation. This suggests that the described anticancer activities associated with curcumin should be further confirmed using a single specific formulation. In this case, there would be real opportunities to consider and use curcumin in future combined cancer therapy.

## 6. Conclusions

The increasing incidence of cancer makes it urgent to find new treatments to cure these very different diseases and improve the quality of life of patients. The main problem with traditional cancer therapies is that the standard drugs cause the frequent occurrence of severe side effects and the development of chemotherapy resistance.

Over time, researchers have considered the use of natural molecules in combination with drugs to enhance the effect of conventional drugs and/or to mitigate the adverse effects that often significantly affect the lives of cancer patients.

Combination therapy has been conceived as a possible solution to multiple drug resistance and the severe adverse effects often associated with the high concentrations of drugs that must be administered to achieve a significant effect. Combining conventional anticancer drugs with natural molecules, the administration of which is not associated with significant adverse effects, could provide a way/solution to use fewer concentrations of each drug. Unfortunately, combination therapy is still a challenge in medical science today, leading researchers to find the best therapeutic response with the least side effects.

Polyphenols represent a popular class of bioactive molecules that are often used in combination with standard drugs. Among them, curcumin is one of the most promising anticancer agents as it combines high biological safety for normal cells/tissues with potent cytotoxic activity against various human cancers.

## Figures and Tables

**Figure 1 ijms-24-12587-f001:**
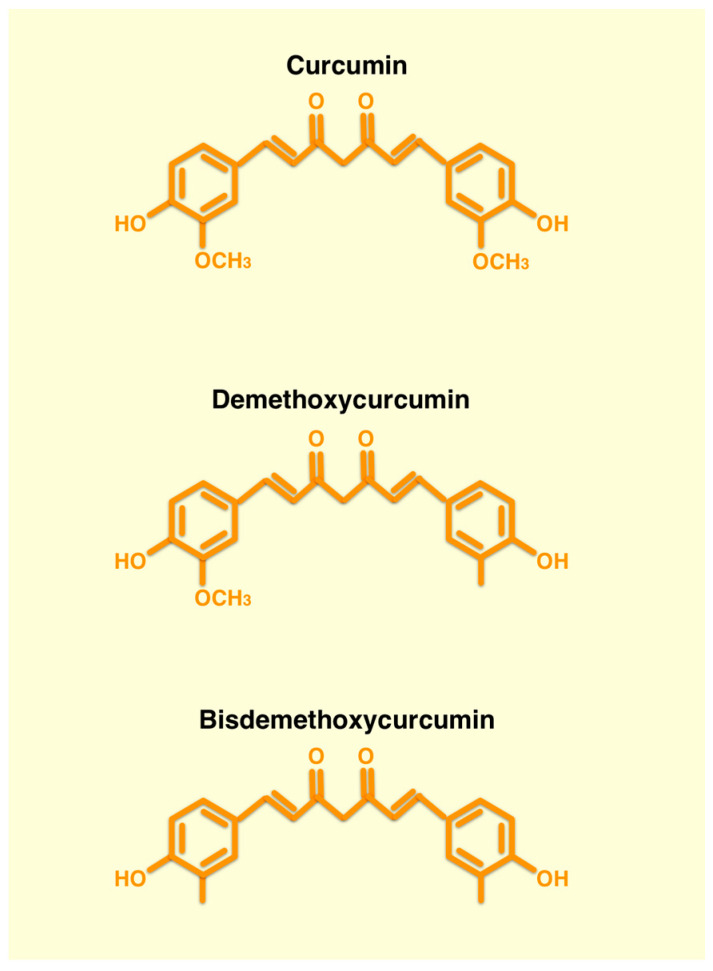
Composition of curcumin extract. Chemical structure of the curcumin and the two curcuminoids contained in the commercial turmeric extracts.

**Figure 2 ijms-24-12587-f002:**
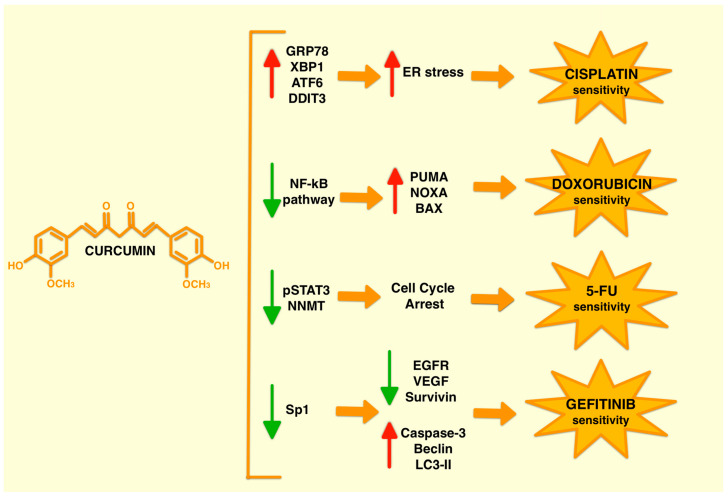
Molecular pathways activated by curcumin in combined treatments. Curcumin combined treatment is able to modulate different signalling pathways involved in inflammation, apoptosis, cell cycle or survival depending on the specific drug used.

**Table 1 ijms-24-12587-t001:** Preclinical studies combining curcumin and standard drugs.

Drug Combined to Curcumin	Tumour	References
Cisplatin 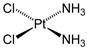	Lung cancer	[66,67]
Ovarian cancer	[68,69]
Breast cancer	[70,71,72,73]
Bladder cancer	[74,75,76]
Glioblastoma	[77,78,79,80]
Laryngeal	[81,82,83]
Doxorubicin 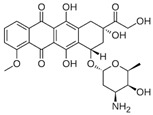	Glioblastoma	[80]
Leukaemia	[84,85,86]
Hepatocellular	[87,88]
Gastric	[89,90]
Neuroblastoma	[91]
5-Fluorouracil 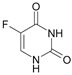	Colorectal	[92,93,94]
Breast	[95]
Liver	[96,97]
Oesophageal	[98,99]
Gastric	[100,101]
Gefitinib 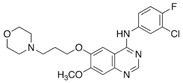	Lung cancer	[102,103]

**Table 2 ijms-24-12587-t002:** Clinical trials combining curcumin and standard drugs.

NCT Number	Study Title	Drug in Combination to Curcumin	Curcumin Used	Condition	Status	Phase	References
NCT02439385	Avastin/FOLFIRI in Combination with Curcumin in Colorectal Cancer Patients With Unresectable Metastasis	FOLFIRI: Folinic acid, 5-FU and Irinotecan	Ginsenoside-modified nanostructured lipid carrier containing Curcumin. Curcumin Sigma-Aldrich	Colorectal Cancer	Completed	Phase 2	[108]
NCT03192059	Study of Pembrolizumab, Radiation and Immune Modulatory Cocktail in Cervical/Uterine Cancer	Cyclophosphamide, Aaspirin, Lansoprazole, Vitamin D, Pembrolizumab, Radiotherapy	Curcumin, CurcuPhyt	Cervical Cancer, Endometrial Cancer, Uterine Cancer	Completed	Phase 2	[109,110]
NCT00192842	Gemcitabine with Curcumin for Pancreatic Cancer	Gemcitabine	Curcumin C3—complex, SabinsaCorporation (Piscataway NJ)	Pancreatic Cancer	Completed	Phase 2	[111]
NCT01490996	Combining Curcumin with FOLFOX Chemotherapy in Patients with Inoperable Colorectal Cancer	FOLFOX: Folinic acid, 5-FU and oxaliplatin	Curcumin C3—complex, SabinsaCorporation (Piscataway NJ)	Colon cancer metastasis	Completed	Phase 1–2	[96,112,113]
NCT01859858	Effect of Curcumin on Dose Limiting Toxicity and Pharmacokinetics of Irinotecan in Patients with Solid Tumors	Irinotecan	Curcumin phosphatidylcholine complex, Meriva	Advanced colorectal cancer	Completed	Phase 1	[110]
NCT00852332	Docetaxel with or without a Phytochemical in Treating Patients With Breast Cancer	Docetaxel	Curcumin capsules	Breast Cancer	Terminated	Phase 2	
NCT02095717	Multicenter Study Comparing Taxotere Plus Curcumin Versus Taxotere Plus Placebo Combination in First-line Treatment of Prostate Cancer Metastatic Castration Resistant (CURTAXEL)	Taxotere	Curcumin formulated in 500 mg capsules	Prostate Cancer Metastatic Castration Resistant	Terminated	Phase 2	
NCT01608139	Study of Curcumin, Vorinostat, and Sorafenib	Vorinostat (SAHA), and Sorafenib	Curcumin powder	Advanced cancers	Withdrawn	Phase 1	
NCT01048983	Reducing Symptom Burden—Non Small Cell Lung Cancer (NSCLC)	Armodafinil, Bupropion, and Minocycline	Curcumin stick pack	Non-Small Cell Lung Cancer	Withdrawn	Phase 1–2	
NCT02321293	A Open-label Prospective Cohort Trial of Curcumin Plus Tyrosine Kinase Inhibitors (TKI) for EGFR-Mutant Advanced NSCLC	Gefitinib and Erlotinib	CurcuVIVA™	Lung cancer	Unknown status	Phase 1	
NCT00295035	Phase III Trial of Gemcitabine, Curcumin and Celebrex in Patients with Metastatic Colon Cancer	CELECOXIB	Curcumin	Colon cancer	Unknown status	Phase 3	
NCT02724202	Curcumin in Combination with 5FU for Colon Cancer	5-flurorouracil	Curcumin BCM-95	Metastatic colon cancer	Unknown status	Early Phase 1	
NCT00486460	Phase III Trial of Gemcitabine, Curcumin and Celebrex in Patients with Advance or Inoperable Pancreatic Cancer	Gemcitabine and Celebrex	Curcumin	Pancreatic Cancer	Unknown status	Phase 3

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
