# Peer review of "An Overview of the Enhanced Effects of Curcumin and Chemotherapeutic Agents in Combined Cancer Treatments"

_ijms, 2023, doi:10.3390/ijms241612587_

Round 1

Reviewer 1 Report

This manuscript provides an overview of the synergetic effects of curcumin in combination with representative chemotherapeutic agents. The authors also show the current situation of preclinical and clinical studies of curcumin-combination therapies. Considering the importance of combination therapies with chemotherapeutic agents in cancer treatments, this manuscript significantly contributes to this field. However, I suggest several things to improve the quality of this manuscript.

<Major points>

1. Although the authors state the synergetic effects of curcumin in combination with many chemotherapeutic agents in the manuscript, these effects are described just in writing. The authors need to present numerical values to show the synergetic effects of curcumin-combination therapies compared to monotherapies. In addition, these contents can be summarized in a Table.

2. A separate section can be inserted between Sections 4 and 5. This new section can include limitations of curcumin-combination therapies, alternatives to those, application strategies, and prospects in more detail, although the authors briefly mention some of those in Section 5.

<Minor points>

1. In Introduction (page 1, line 41), what does the dot in 609.360 indicate? Is that a clerical error for a comma?

2. A subsection title can be given to the part before Section 2.1 (page 2, line 91 - page 4, line 153). 

3. The seven paragraphs (page 4, lines 132-153) can be integrated appropriately and partly. 

4. The initialism ‘NSCLC’ should be located after the full-length word (page 5, line 181).

5. In Table 1, chemical structures of the respective drugs can be added to the table. In addition, types of cancer should be phrased consistently.

6. Section 3 can be divided into subsections with the respective drugs combined with curcumin.

7. in Section 3, many references are missing (e.g. page 10, lines 423-426, etc.).

8. In Table 2, references should be summarized in a separate column. In addition, some NCT numbers do not have references. If references cannot be found, please state it in the manuscript.

Minor editing of English language required

Reviewer 2 Report

This manuscript provided a comprehensive review of using curcumin to combine with traditional chemotherapeutics for cancer therapy. This manuscript was well-written and well-organized. The readers should be benefited from this review regarding Curcumin's pharmacological actions and the potential application of the Curcumin-chemotherapeutics combination in cancer therapy. Comments on this manuscript are provided as follows.

1. The title implied a synergy in cytotoxicity using curcumin-chemotherapeutics combination is generally observed in all studies. However, rather than proclaiming synergy, most of the references cited in this manuscript stated an enhancement of drug efficacy or circumventing drug resistance when combined with curcumin. Accordingly, the "synergistic effect" in the title is argumentative and thus must be revised to avoid misleading the readers.

2. Including reports published in the year 2023 about the curcumin-chemotherapeutics combination in cancer therapy is highly recommended.

3. Before the conclusion section, please include a paragraph stating the benefits and limitations of using curcumin for combined cancer therapy.

4. The font of the "k" in NF-kB should be "symbol."

5. Several grammar mistakes in this manuscript could be improved. Please check the grammar carefully.

Several grammar mistakes in this manuscript could be improved. Please check the grammar carefully.

Reviewer 3 Report

In recent years,  cancer has been a scourge of modern society that is global and widespread. All this determines the relevance of the problem under consideration and the significance of the results presented by the authors. This paper presented the study of the synergistic effects of curcumin, a polyphenol that has been shown to have extensive cytotoxic functions against cancer cells, including combined treatment. The advantage of combination treatments is the possibility of increasing the efficacy of drugs and reducing their dose, which leads to a reduction in adverse effects. This is a very interesting, useful and up-to-date article.

The literature cited in this paper is sufficient in volume and content (110). All 2 figures are of sufficiently high quality. The tables are well presented. I recommend that the journal editors accept the article with minor grammatical and technically corrections.

Dear Author!

 Please revisit the article with reg. no.2537525  in accordance with Editor’s comments:

·        The literature should be edited according to the requirements of the journal.

Round 2

Reviewer 1 Report

The manuscript has been quite improved.

Reviewer 2 Report

No more question to add